# Multi-Operator Few-Shot Learning for Generalization Across PDE Families

## Abstract

Learning solution operators for partial differential equations (PDEs) has become a foundational task in scientific machine learning. However, existing neural operator methods require abundant training data for each specific PDE and lack the ability to generalize across PDE families. In this work, we propose MOFS: a unified multimodal framework for **m**ulti-**o**perator **f**ew-**s**hot learning, which aims to generalize to unseen PDE operators using only a few demonstration examples. Our method integrates three key components: (i) multi-task self-supervised pre-training of a shared Fourier Neural Operator (FNO) encoder to reconstruct masked spatial fields and predict frequency spectra, (ii) text-conditioned operator embeddings derived from statistical summaries of input-output fields, and (iii) memory-augmented multimodal prompting with gated fusion and cross-modal gradient-based attention. We adopt a two-stage training paradigm that first learns prompt-conditioned inference on seen operators and then applies end-to-end contrastive fine-tuning to align latent representations across vision, frequency, and text modalities. Experiments on PDE benchmarks, including Darcy Flow and Incompressible Navier Stokes variants, demonstrate that our model outperforms existing operator learning baselines in few-shot generalization. Extensive ablations validate the contributions of each modality and training component. Our approach offers a new foundation for universal and data-efficient operator learning across scientific domains.

## 1 Introduction

Learning solution operators for partial differential equations (PDEs) has become a central task in scientific machine learning, enabling data-driven modeling of complex physical systems across domains such as fluid dynamics, materials science, and climate simulation. Neural operator methods, particularly the FNO (Li et al., 2020), have shown strong performance in approximating PDE solution maps by learning continuous mappings between function spaces. However, existing operator learning frameworks typically require large amounts of training data for each specific PDE and lack the ability to generalize across different PDE families that vary in governing equations, boundary conditions, or parameter regimes. To overcome this limitation, we study the problem of multi-operator few-shot learning across PDE families: given only a small set of input-output demonstrations from an unseen PDE operator, the goal is to accurately predict solutions for new query inputs. This formulation presents two fundamental challenges: (i) how to extract structural priors that are transferable across heterogeneous PDEs, and (ii) how to efficiently leverage limited demonstrations for generalization to novel operators. We propose a new multimodal framework MOFS for **m**ulti-**o**perator **f**ew-**s**hot learning, which integrates spatial-frequency pretraining, semantic textual embeddings, and memory-augmented multimodal inference. Our approach integrates three key components:

- Self-supervised spatial-frequency pretraining: We introduce a multi-task pretraining scheme on a shared FNO encoder that jointly reconstructs masked spatial fields and predicts their spectral representations, thereby capturing generalizable structural and frequency-domain priors.

- Text-conditioned operator embedding: To leverage semantic information about PDE datasets, we generate descriptive natural language summaries based on field statistics such

as means, ranges, gradient magnitudes, which are embedded using a pretrained BERT model (Devlin et al., 2019). These embeddings serve as operator-level priors and are fused with vision and frequency features via cross-attention.

- Memory-augmented multimodal prompting: We maintain a dynamic memory buffer of prior PDE examples, from which relevant prompts are retrieved using similarity-based attention. A prompt-conditioned decoder then predicts the solution using cross-modal gradient-based attention and soft prompt embeddings.

To enable robust generalization, we adopt a two-stage training strategy: (i) supervised few-shot learning using prompt-conditioned loss on seen operators, and (ii) end-to-end contrastive fine-tuning to align latent representations across inputs, outputs, and text descriptions. Our framework is evaluated on PDE benchmarks, including Darcy Flow and Incompressible Navier Stokes. Results show that our model outperforms prior neural operator methods in the few-shot setting and demonstrates cross-operator generalization capabilities. Our contributions can be summarized as follows:

- To our knowledge, MOFS is the first multi-operator few-shot operator learning framework that integrates frequency-aware self-supervision, semantic text conditioning, and memory-augmented multimodal prompting.

- We design a contrastive fine-tuning strategy to align cross-modal and operator-level representations for robust generalization to unseen PDEs.

- We demonstrate superior performance on benchmark PDE datasets, validating the effectiveness of our approach for rapid operator adaptation with few demonstrations.

## 2 RELATED WORK

**Neural Operators.** Neural operators aim to learn mappings between infinite-dimensional function spaces, enabling solution generalization across varying PDE input conditions. Notable examples include the DeepONet (Lu et al., 2019), which approximates nonlinear operators using branch and trunk networks, and the FNO (Li et al., 2020), which captures global dependencies via spectral convolutions. Extensions of these models, such as Galerkin-based (Cao, 2021) and attention-based (Guibas et al., 2021; Pfaff et al., 2020) variants, have improved accuracy and interpretability, particularly in high-dimensional and irregular domains.

**Few-Shot and Meta-Learning for Operators.** Few-shot learning has been extensively studied in the context of computer vision (Vinyals et al., 2016) and natural language processing (Brown et al., 2020). Recent works extend this paradigm to operator learning, such as Meta-PDE (Qin et al., 2022) and Lemon (Sun et al., 2024), which leverage meta-learning to generalize across PDE families. These approaches typically require either episodic training or adaptive parameter generation mechanisms, but often struggle to integrate heterogeneous input modalities or exploit structural physics priors effectively.

**Multimodal and Memory-Augmented Architectures.** Combining visual and textual modalities has gained prominence in few-shot learning through architectures like CLIP (Radford et al., 2021), BLIP (Li et al., 2022), and Flamingo (Alayrac et al., 2022). These models utilize cross-modal contrastive or generative objectives to align embeddings from different modalities. Recent advances like STRAP (Memmel et al., 2024) and CoCoOp (Zhou et al., 2022) further explore memory-augmented retrieval and prompt-tuning techniques for in-context learning. Our model extends these ideas to the operator domain by introducing gated fusion and attention-based memory for PDE tasks.

**Self-Supervised and Contrastive Learning.** Self-supervised pretraining via masked prediction (He et al., 2022; Bao et al., 2021) has shown to enhance generalization in sparse-data regimes. Contrastive learning (Chen et al., 2020; He et al., 2020) has been used to align multimodal representations, improve robustness, and learn semantic structure without supervision. In operator learning, contrastive objectives remain underexplored. Our framework bridges this gap by incorporating physics-consistent and modality-aligned contrastive losses in both supervised and unsupervised stages.

## 3 PROBLEM FORMULATION

Let $\{\mathcal{O}_k\}_{k=1}^K$ be a collection of $K$ distinct PDE operators, where each $\mathcal{O}_k$ defines a nonlinear mapping $\mathcal{G}_k : \mathcal{A}_k \to \mathcal{U}_k$ between input fields $a \in \mathcal{A}_k \subset \mathbb{R}^{H \times W}$ and output fields $u \in \mathcal{U}_k \subset \mathbb{R}^{H \times W}$. Each operator $\mathcal{O}_k$ provides a dataset

$$\mathcal{D}_k = \{(a_j^{(k)}, u_j^{(k)})\}_{j=1}^J, \quad u_j^{(k)} = \mathcal{G}_k(a_j^{(k)}),$$

where $N$ is the number of samples, $J$ is the length of prompt. We split $\{\mathcal{O}_k\}_{k=1}^K$ into a set of source operators $\mathcal{O}_{\text{train}} = \{\mathcal{O}_k\}_{k \in \mathcal{I}_{\text{train}}}$ and a previously unseen PDE dataset $\mathcal{O}_{\text{test}} = \mathcal{O}_{\mathcal{I}_{\text{test}}}$ with $\mathcal{I}_{\text{test}} \notin \mathcal{I}_{\text{train}}$. The model is trained on the set of source operators and evaluated on the test operator with only a few input-output examples

$$\mathcal{D}_{\text{test}} = \{(a_j^{(\text{test})}, u_j^{(\text{test})})\}_{j=1}^J.$$

The objective is to predict the output $u_*^{(\text{test})} = \mathcal{G}_{\text{test}}(a_*^{(\text{test})})$ for a new query input $a_*^{(\text{test})}$. The goal is to minimize the empirical prediction loss across operators

$$\min_\theta \ \mathbb{E}_{\mathcal{O}_k \sim \mathcal{O}} \left[ \mathbb{E}_{(a_*, u_*) \sim \mathcal{D}_k} \ \mathcal{L}_2(f_\theta(a_*; \mathcal{D}_k), u_*) \right]. \tag{1}$$

The ultimate objective is to learn an operator network $f_\theta$ that can generalize to unseen PDE operators given only a small number of demonstration examples.

## 4 METHODOLOGY

### 4.1 MULTI-TASK PRETRAINING WITH SPATIAL AND FREQUENCY RECONSTRUCTION

To enable effective few-shot generalization across PDE families, we pretrain a shared encoder using a multi-task self-supervised objective. Specifically, the model is trained to jointly reconstruct masked spatial input fields and predict their corresponding frequency spectra.

**Input Representation.** We preprocess and normalize $a, u$ using a Gaussian normalizer, then randomly mask a portion of its spatial content $\tilde{a} = a \odot M_a$, $\tilde{u} = u \odot M_u$ with $M_a, M_u \sim \text{Bernoulli}(1 - \rho)$, where $M_a, M_u \in \{0, 1\}^{H \times W}$ is a binary mask with masking ratio $\rho$, and $\odot$ denotes element-wise multiplication.

**FNO Encoder.** We employ a FNO encoder $\mathcal{E}_\phi : \mathbb{R}^{1 \times H \times W} \to \mathbb{R}^{d \times H \times W}$ that lifts the input to a high-dimensional latent space via spectral convolution layers $f_a = \mathcal{E}_\phi(\tilde{a})$, $f_u = \mathcal{E}_\phi(\tilde{u})$, $f = \text{Concat}(f_a, f_u)$.

**Spatial Decoder.** The spatial decoder $\psi_a$ and $\psi_u$ predict the original values from masked latent representations $\hat{a} = \psi_a(f)$, $\hat{u} = \psi_u(f)$ We supervise this decoder using a relative $L_2$ loss computed over the masked regions

$$\mathcal{L}_{\text{spatial}} = \frac{\|(1 - M_a) \odot (\hat{a} - a)\|_2^2}{\|(1 - M_a) \odot a\|_2^2 + \varepsilon} + \frac{\|(1 - M_u) \odot (\hat{u} - u)\|_2^2}{\|(1 - M_u) \odot u\|_2^2 + \varepsilon} \tag{2}$$

where $\|\cdot\|_2$ denotes the Frobenius norm, and $\varepsilon > 0$ is a small constant added for numerical stability.

**Frequency Spectrum Decoder.** To enhance spectral understanding, we introduce another decoder $D_\Phi$ that predicts the magnitude of the 2D Fourier transform of the original input by $\hat{F} = \mathcal{D}_\Phi(f)$, and we use 2D Fast Fourier Transform (FFT) as ground truth $\tilde{F} \approx |\mathcal{F}(a)|$, where $\mathcal{F}(\cdot)$ denotes 2D FFT and $|\cdot|$ denotes the magnitude. The frequency loss is computed using the relative $L_2$ loss

$$\mathcal{L}_{\text{freq}} = \frac{\|\hat{F} - |\mathcal{F}(a)|\|_2}{\||\mathcal{F}(a)|\|_2}. \tag{3}$$

The final multi-task pretraining loss combines both objectives $\mathcal{L}_{\text{pretrain}} = \mathcal{L}_{\text{spatial}} + \alpha \cdot \mathcal{L}_{\text{freq}}$, where $\alpha$ balances the two tasks.

## 4.2 Text-Guided Embedding for PDE Operators

In order to incorporate semantic prior knowledge into PDE operator learning, we construct a text-based embedding module that converts structured PDE data into natural language descriptions and encodes them using a pretrained language model. Let $\mathcal{T}$ denote a tokenizer and $\mathcal{B}$ be a pretrained BERT model (Devlin et al., 2019) with hidden size $d_{\text{bert}}$. We define a projection head $W_p \in \mathbb{R}^{d_{\text{bert}} \times d}$ to map BERT representations to an embedding space $\mathbb{R}^d$. The details are as follows.

**Descriptive Text Generation for PDE Datasets.** Each PDE dataset consists of samples $\{(a_i, u_i)\}_{i=1}^N$, where $a_i$ denotes the input coefficient field and $u_i$ represents the corresponding PDE solution. For $k$-th PDE dataset, we generate a natural language description $\mathcal{T}_k$ using the following statistical attributes: (i) Mean, standard deviation, minimum, and maximum of $a_i$ and $u_i$; (ii) Gradient magnitude statistics of $u_i$ to quantify spatial smoothness. The descriptive sentence $\mathcal{T}_k$ for each sample $(a_i^{(k)}, u_i^{(k)})$ is

> *"This is a* `<dataset_name>` *PDE sample. The input coefficient field has mean $\mu_a$, std $\sigma_a$, min $a_{\min}$, and max $a_{\max}$. The output solution field has mean $\mu_u$, std $\sigma_u$, range $[u_{\min}, u_{\max}]$, and gradient magnitude mean $\mu_{\nabla u}$ with std $\sigma_{\nabla u}$."*

**Text Encoder.** For $k$-th PDE, given a batch of tokenized descriptions $(x, b)$, where $x \in \mathbb{N}^{B \times L}$ denotes input token IDs, $b \in \{0, 1\}^{B \times L}$ is the corresponding attention mask and $L$ is the maximum length of the text, the encoder computes

$$H_t^{(k)} = \mathcal{B}(x, b) \in \mathbb{R}^{B \times L \times d_{\text{bert}}}, \quad z_t^{(k)} = \frac{1}{L}\sum_{i=1}^{L} H_{t:,i,:}^{(k)} \in \mathbb{R}^{B \times d_{\text{bert}}},$$

$$e^{(k)} = z_t^{(k)} \cdot W_p \in \mathbb{R}^{B \times d}, \quad \bar{e}^{(k)} = \frac{1}{B}\sum_{i=1}^{B} e_{i,:}^{(k)} \in \mathbb{R}^d. \tag{4}$$

This aggregated vector $\bar{e}^{(k)}$ serves as a semantic embedding of the $k$-th PDE operator, capturing both physical statistics and linguistic priors derived from the underlying dataset. We aim to obtain a text embedding for each PDE operator in this step.

## 4.3 Multimodal Fusion and Decoder

This module extracts high-level visual features from input and output fields via a pretrained image encoder and fuses them with spectral encodings from FNO Encoder and text embeddings from BERT.

**FNO Encoder.** We encode $a$ and $u$ using FNO encoder trained in the pre-traning step to capture frequency representations with positional encoding $P \in \mathbb{R}^{1 \times d \times H \times W}$ which is a learnable parameter by

$$f_a = \mathcal{E}_\phi(a), \quad f_u = \mathcal{E}_\phi(u), \quad f_a^{\text{pos}} = f_a + P, \quad f_u^{\text{pos}} = f_u + P. \tag{5}$$

**Vision Encoder.** A pretrained ResNet-18 is used to encode scalar fields by $\phi_{\text{vis}}(x) = \text{ResNet18}(x)$. We encode $a$ and $u$ repectively to capture geometric and high-level spatial patterns

$$v_a = \phi_{\text{vis}}(a), \quad v_u = \phi_{\text{vis}}(u). \tag{6}$$

**Fusion via Gating.** The frequency and vision features are adaptively fused using learned gating

$$f_a^{\text{cat}} = \text{Concat}[f_a^{\text{pos}}, v_a]), \quad f_u^{\text{cat}} = \text{Concat}[f_u^{\text{pos}}, v_u])$$
$$\tilde{f}_a = \sigma(W_a f_a^{\text{cat}}) \cdot f_a^{\text{pos}} + (1 - \sigma(W_a f_a^{\text{cat}})) \cdot v_a,$$
$$\tilde{f}_u = \sigma(W_u f_u^{\text{cat}}) \cdot f_u^{\text{pos}} + (1 - \sigma(W_u f_u^{\text{cat}})) \cdot v_u, \tag{7}$$

where $W_a, W_u \in \mathbb{R}^{1 \times 2d}$.

**Fusion with text.** We fuse textual embedding $\bar{e}^{(k)} \in \mathbb{R}^d$ with $\tilde{f}_a$ via cross-attention $\phi_c$ by

$$\tilde{f}_{\text{fused}} = \phi_c(\tilde{f}_a, \bar{e}^{(k)}, \bar{e}^{(k)}). \tag{8}$$

where $\phi_c$ is defined as

$$\phi_c(Q, K, V) = \text{Concat}(\text{head}_1, \ldots, \text{head}_h)W^O \tag{9}$$

$$\text{head}_i = \text{Softmax}\left(\frac{Q_i K_i^\top}{\sqrt{d_h}}\right) V_i, \quad \text{for } i = 1, \ldots, h,$$

$$Q_i = QW_i^Q, \quad K_i = KW_i^K, \quad V_i = VW_i^V,$$

where $W_i^Q, W_i^K, W_i^V \in \mathbb{R}^{d \times d_h}$, $W^O \in \mathbb{R}^{d \times d}$, $d_h = d/h$ is the head dimension. Then we fuse $\tilde{f}_a$ and $\tilde{f}_{\text{fused}}$ by

$$\tilde{f}'_a = \gamma(\tilde{f}_{\text{fused}}) \cdot \tilde{f}_a + \beta(\tilde{f}_{\text{fused}}) \tag{10}$$

where $\gamma$ and $\beta$ are Linear layers. Then the output $\tilde{f}'_a$ goes through a self-attention and we get $\tilde{f}''_a$ by

$$\tilde{f}''_a = \text{LN}(Z_1 + \text{MLP}(Z_1)) \tag{11}$$

where $Z_1 = \text{LN}(\tilde{f}'_a + \phi_c(\tilde{f}'_a))$, LN denotes LayerNorm. Then the output $\tilde{f}''_a$ comes to a gradient-based cross attention $\phi'_c$ and we get $\hat{f}_a = \phi'_c(\tilde{f}''_a, \tilde{f}_u, \tilde{f}_u)$, where $\phi'_c$ is defined

$$\phi'_c(Q, K, V) = \text{LN}(\tilde{Q} - \eta \cdot \text{MLP}(H)), \quad H = \text{Softmax}\left(\frac{\tilde{Q}\tilde{K}^\top}{\sqrt{d}}\right)\tilde{V}, \tag{12}$$

$$\tilde{Q} = \text{LN}(Q), \tilde{K} = \text{LN}(K), \tilde{V} = \text{LN}(V),$$

where $\eta$ is a learnable step size.

**Memory Buffer.** We incorporate a memory buffer $\mathcal{M} = \{(\tilde{f}_a, \tilde{f}_u))\}$ with key-value pairs of previously seen PDE features.

**Query Encoding.** We apply encoder defined in Eq 5, 6 and 7 to query and get $\tilde{f}_q$ by

$$\tilde{f}_q = \sigma(W_q f_q^{\text{cat}}) \cdot f_q^{\text{pos}} + (1 - \sigma(W_q f_q^{\text{cat}})) \cdot v_q, \tag{13}$$

$$f_q^{\text{pos}} = \mathcal{E}_\phi(a_q) + P, \quad v_q = \phi_{\text{vis}}(a_q), \quad f_q^{\text{cat}} = \text{Concat}[f_q^{\text{pos}}, v_q].$$

Query representations $\tilde{f}_q$ retrieve the top-$k$ relevant memory vectors $\{(\tilde{f}_a^{(i)}, \tilde{f}_u^{(i)}\}_{i=1}^k$ from memory buffer $\mathcal{M}$ by similarity. Then based on these selected pairs, we get weights $w$ and a weighted key-value pair $(\hat{z}_a, \hat{z}_u)$ by

$$\hat{z}_a = \sum_{i=1}^k w_i \cdot \tilde{f}_a^{(i)}, \quad \hat{z}_u = \sum_{i=1}^k w_i \cdot \tilde{f}_u^{(i)}, \quad f_q^{\text{proj}} = W'_q \tilde{f}_q, \tag{14}$$

$$w_i = \frac{\exp\left(\left(\langle f_q^{\text{proj}}, \tilde{f}_a^{(i)}\rangle / \tau + \alpha \cdot l_i\right)\right)}{\sum_j \exp\left(\left(\langle f_q^{\text{proj}}, \tilde{f}_a^{(j)}\rangle / \tau + \alpha \cdot l_j\right)\right)}. \tag{15}$$

where $l_i$ is the quality score of memory key $\tilde{f}_a^{(i)}$, defined as $l_i = e^{-\mathcal{L}_2(\hat{u}^{(i)}, u^{(i)})}$, and $\alpha$ is a quality scaling factor, $\tau$ is a temperature. Then we combine fusion vector $\hat{f}_a$ with weighted memory key $\hat{z}_a$ and get final fusion vector $\hat{f}'_a$ which is the input to the decoder by

$$\hat{f}'_a = \hat{f}_a + \sigma(W_m f_m^{\text{cat}}) \cdot \hat{z}_a, \quad f_m^{\text{cat}} = \text{Concat}[\hat{f}_a, \hat{z}_a] \tag{16}$$

Then for query encoding $\tilde{f}_q$ we add an operator ID embedding and get $\tilde{f}'_q$ by

$$\tilde{f}'_q = \tilde{f}_q + e_o^{(k)} \tag{17}$$

where $e_o^{(k)}$ is a learnable embedding layer for operator ID $k$. Then we put $\tilde{f}'_q$ to a cross attention network $\phi_c$ with weighted memory value $\hat{z}_u$ and get updated query encoding $\tilde{f}''_q$, then we combine $\tilde{f}''_q$ with weighted memory value $\hat{z}_u$ to get the final query embedding $\hat{f}_q$ by

$$\tilde{f}''_q = \tilde{f}'_q + \phi_c(\tilde{f}'_q, \hat{z}_u), \quad \hat{f}_q = \gamma(\hat{z}_u) \odot \tilde{f}''_q + \beta(\hat{z}_u). \tag{18}$$

**Decoder.** Each operator is associated with a learnable soft prompt $P_{\text{soft}}^{(k)} \in \mathbb{R}^{L \times d}$ where $L$ is the length of soft prompt. The prompt-conditioned features are passed to the decoder $\mathcal{D}_\gamma$ by

$$\hat{u}_q = \mathcal{D}_\gamma \left( \{(a_j, u_j)\}_{j=1}^J, a_q, P_{\text{soft}}^{(k)} \right).$$

Specifically, we use gradient-based cross attention $\phi_c'$ defined in Eq 12 to get prediction of $u_q$ by

$$\hat{u}_q = \text{MLP} \left( \phi_c' \left( \text{Concat} \left( \gamma(\hat{f}_a'), P_{\text{soft}}^{(k)} \right), \beta(\hat{f}_q) \right) \right) \tag{19}$$

## 4.4 Training Loop

We propose a two-stage training procedure for learning a universal operator model capable of generalizing to unseen PDE families via few-shot prompting and contrastive reasoning.

**Prompt-Conditioned Supervised Learning.** In the first stage, we freeze most layers of the encoder $\mathcal{E}_\phi$ and train the other parts of the model on few-shot prompts from training PDE families. The loss function is

$$\mathcal{L}_{\text{stage1}} = \mathcal{L}_2(\hat{u}, u) = \frac{\|u - \hat{u}\|_2}{\|u\|_2}. \tag{20}$$

**Supervised contrastive loss.** Let $\{(z_i^{(a)}, z_i^{(u)}, y_i)\}_{i=1}^B$ denote a minibatch of $B$ examples, where $z_i^{(a)} \in \mathbb{R}^d$ and $z_i^{(u)} \in \mathbb{R}^d$ are the $d$-dimensional latent representations of the input and output fields for the $i$-th sample, and $y_i \in \mathcal{Y}$ denotes the associated operator ID label. We define a symmetric similarity matrix $S \in \mathbb{R}^{B \times B}$ where each entry is the temperature-scaled cosine similarity

$$S_{ij} = \frac{\langle \hat{z}_i^{(a)}, \hat{z}_j^{(u)} \rangle}{\tau}, \quad \text{with} \quad \hat{z} = \frac{z}{\|z\|_2}, \quad \tau > 0. \tag{21}$$

The diagonal entries of $S$ are suppressed to avoid self-comparisons such that $S_{ij} = \begin{cases} S_{ij} & \text{if } i \neq j, \\ -\infty & \text{if } i = j. \end{cases}$

To enable progressive learning, we introduce a difficulty factor $\lambda_t = \min \left( 1.0, \frac{t}{0.3T} \right)$ that increases with epoch $t$ over the total training epochs $T$. We define a hard negative threshold $\theta_t = 0.7 - 0.3\lambda_t$ and classify negatives with $S_{ij} > \theta_t$ as hard negatives.

$$\mathcal{L}_c = -\frac{1}{B} \sum_{i=1}^B \log \frac{\sum\limits_{j : y_j = y_i} \exp(S_{ij})}{\sum\limits_{j \neq i} \exp(S_{ij}) + \lambda_t \sum\limits_{j \in \mathcal{H}_i} \exp(S_{ij})} \tag{22}$$

where $\mathcal{H}_i = \{j \neq i \mid y_j \neq y_i, \ S_{ij} > \theta_t\}$ denotes the set of hard negatives for sample $i$. This progressive curriculum facilitates learning by starting with easy negatives and gradually introducing harder examples, improving generalization across operators.

**End-to-End Contrastive Fine-Tuning.** In the second stage, we unfreeze all model parameters and fine-tune the whole model using a hybrid loss that incorporates both physics and contrastive supervision. We use contrastive loss function $\mathcal{L}_c(\cdot, \cdot)$ defined in Eq 22 to align consistency of $(z_a, z_u)$, $(z_u, \bar{e}^{(k)})$, $(z_a, \hat{z}_u)$ and introduce consistency loss by

$$\mathcal{L}_{\text{consis}} = \lambda_1 \cdot \mathcal{L}_c(z_a, z_u) + \lambda_2 \cdot \mathcal{L}_c(z_u, e^{(k)}) + \lambda_3 \cdot \mathcal{L}_c(z_a, \hat{z}_u). \tag{23}$$

In order to encourages the memory embeddings to be diverse, we introduce $\mathcal{L}_{\text{diversity}}$ which averages squared similarity between all distinct key pairs in memory buffer by

$$\mathcal{L}_{\text{diversity}} = \lambda_4 \cdot \frac{2}{N(N-1)} \sum_{i=1}^N \sum_{j=i+1}^N \left( \tilde{f}_a^{(i)\top} \tilde{f}_a^{(j)} \right)^2 \tag{24}$$

where $N$ is the number of keys in memory buffer. Then the training loss function in the second stage is defined as

$$\mathcal{L}_{\text{stage2}} = \mathcal{L}_2(\hat{u}, u) + \mathcal{L}_{\text{consis}} + \mathcal{L}_{\text{diversity}}. \tag{25}$$

The model is trained end-to-end to jointly improve few-shot prediction accuracy and cross-modal generalization. The whole training logic is shown in Algorithm 1, 2, 3. The work flow of MOFS is presented in Figure 1.

# 5 EXPERIMENTS

---

**Algorithm 1** Pre-train Stage: Multi-Task Self-Supervised Learning

---

**Require:** Multi-operator datasets $\{\mathcal{D}_k\}_{k=1}^K$, masking rate $\rho$, frequency weight $\alpha$
1: **for all** $(a, u) \in \cup_k \mathcal{D}_k$ **do**
2:     Sample masks: $M_a, M_u \sim \text{Bernoulli}(1-\rho)$
3:     Apply masks: $\tilde{a} \leftarrow a \odot M_a, \quad \tilde{u} \leftarrow u \odot M_u$
4:     Encode: $f_a \leftarrow \mathcal{E}_\phi(\tilde{a}), \quad f_u \leftarrow \mathcal{E}_\phi(\tilde{u})$
5:     Concatenate: $f \leftarrow [f_a; f_u]$
6:     Decode: $\hat{a} \leftarrow \psi_a(f), \quad \hat{u} \leftarrow \psi_u(f)$
7:     Compute spatial loss $\mathcal{L}_{\text{spatial}}$ using Eq. 2
8:     Predict FFT: $\hat{F} \leftarrow \mathcal{D}_\Phi(f), \quad \tilde{F} \leftarrow |\mathcal{F}(a)|$
9:     Compute frequency loss $\mathcal{L}_{\text{freq}}$ using Eq. 3
10:    Total loss: $\mathcal{L}_{\text{pretrain}} = \mathcal{L}_{\text{spatial}} + \alpha \cdot \mathcal{L}_{\text{freq}}$
11:    Update $\mathcal{E}_\phi, \psi_a, \psi_u, \mathcal{D}_\Phi$ via gradient descent
12: **end for**

---

**Algorithm 2** Stage 1: Prompt-Conditioned Few-Shot Supervised Learning

---

**Require:** Few-shot training dataset $\{\mathcal{D}_k\}_{k=1}^K$, each with prompts $\{(a_j, u_j)\}_{j=1}^J$; Query input-output pair $(a^*, u^*)$; Soft prompt $P_{\text{soft}}^{(k)}$; Memory buffer $\mathcal{M}$; FNO encoder $\mathcal{E}_\phi$, Text embedding map $\bar{e}$
1: **for all** $(a^*, u^*) \in \cup_k \mathcal{D}_k$ **do**
2:     Encode inputs: $f_a^* \leftarrow \mathcal{E}_\phi(a^*), \quad f_u^* \leftarrow \mathcal{E}_\phi(u^*)$
3:     Add $(f_a^*, f_u^*)$ pairs into memory buffer $\mathcal{M}$
4:     Retrieve prompts $\{(a_j, u_j)\}_{j=1}^J$ for operator $k$
5:     Fuse prompt and query: $f_{\text{prompt}} \leftarrow \text{GradientCrossAttention}(f_a^*, f_u^*, \{a_j, u_j\}, \bar{e}^{(k)})$
6:     Decode prediction: $\hat{u}^* \leftarrow D_\gamma(f_{\text{prompt}}, P_{\text{soft}}^{(k)})$
7:     Compute loss: $\mathcal{L}_{\text{stage1}} = \mathcal{L}_2(\hat{u}^*, u^*)$
8:     Update all parameters except FNO encoder $\mathcal{E}_\phi$ via gradient descent
9: **end for**

---

**Algorithm 3** Stage 2: End-to-End Contrastive Fine-Tuning

---

**Require:** All model parameters; Few-shot training dataset $\{\mathcal{D}_k\}_{k=1}^K$, each with prompts $\{(a_j, u_j)\}_{j=1}^J$; Query input-output $(a^*, u^*)$; Soft prompt $P_{\text{soft}}^{(k)}$; Memory buffer $\mathcal{M}$; Text embedding map $\bar{e}$; Supervised contrastive loss $\mathcal{L}_c$, Hyperparameters $\lambda_1, \lambda_2, \lambda_3$
1: **for all** $(a^*, u^*) \in \cup_k \mathcal{D}_k$ **do**
2:     Encode: $z_a \leftarrow \mathcal{E}_\phi(a^*), \quad z_u \leftarrow \mathcal{E}_\phi(u^*)$
3:     Add $(z_a, z_u)$ pairs into memory buffer $\mathcal{M}$
4:     Retrieve prompts $\{(a_j, u_j)\}_{j=1}^J$ for operator $k$
5:     Fuse prompt and query: $f_{\text{prompt}} \leftarrow \text{GradientCrossAttention}(z_a, z_u, \{a_j, u_j\}, \bar{e}^{(k)})$
6:     Decode prediction: $\hat{u}^* \leftarrow D_\gamma(f_{\text{prompt}}, P_{\text{soft}}^{(k)})$
7:     Compute prediction loss: $\mathcal{L}_{\text{pred}} \leftarrow \mathcal{L}_2(\hat{u}^*, u^*)$
8:     Retrieve from memory: $(\hat{z}_a, \hat{z}_u) \leftarrow \mathcal{M}(a^*)$
9:     Compute consistency loss: $\mathcal{L}_{\text{consis}} = \lambda_1 \cdot \mathcal{L}_c(z_a, z_u) + \lambda_2 \cdot \mathcal{L}_c(z_u, e^{(k)}) + \lambda_3 \cdot \mathcal{L}_c(z_a, \hat{z}_u)$
10:    Compute diversity loss $\mathcal{L}_{\text{diversity}}$ using keys in buffer
11:    Total loss: $\mathcal{L}_{\text{stage2}} = \mathcal{L}_{\text{pred}} + \mathcal{L}_{\text{consis}} + \mathcal{L}_{\text{diversity}}$
12:    Update all parameters via gradient descent
13: **end for**

---

We conduct extensive experiments to evaluate the effectiveness of our proposed model in the context of multi-operator few-shot learning across PDE families. We have access to 11 distinct PDE operator datasets, each representing a different family of PDEs (Takamoto et al., 2022). There are 5 Darcy Flow variants with varying permeability contrasts $\beta \in \{0.01, 0.1, 1.0, 10.0, 100.0\}$ and 6 Incompressible Navier Stokes variants with variations in initial conditions $\in \{0, 1, 10, 100, 101, 102\}$. Each dataset contains input-output pairs $(a, u)$, where $a$ is the coefficient field and $u$ is the PDE solution. Each dataset is treated as a distinct operator $\mathcal{O}_k$, and we simulate few-shot generalization by selecting $J = 4$ input-output demonstration pairs per test query. To evaluate the model's ability to generalize to unseen PDE operators, in each experiment, we designate 1 operator dataset as the test family, and use the remaining 10 operator datasets for training. For each training operator, we sample 10 data points. For the test operator, we sample 10 examples to simulate few-shot inference. We repeat this process by rotating the test operator across all 11 PDE datasets. We compare against standard operator learning models FNO (Li et al., 2020), DeepONet (Lu et al., 2019) and UNet (Ronneberger et al., 2015). To assess the impact of each modality and training component, we perform the following ablations: (i) w/o vision: retain only FNO encoder. (ii) w/o text: remove text embeddings from the fusion module. (iii) w/o memory: disable memory retrieval from external operator memory buffer. (iv) w/o pre-train stage: skip self-supervised spatial-frequency pretraining.

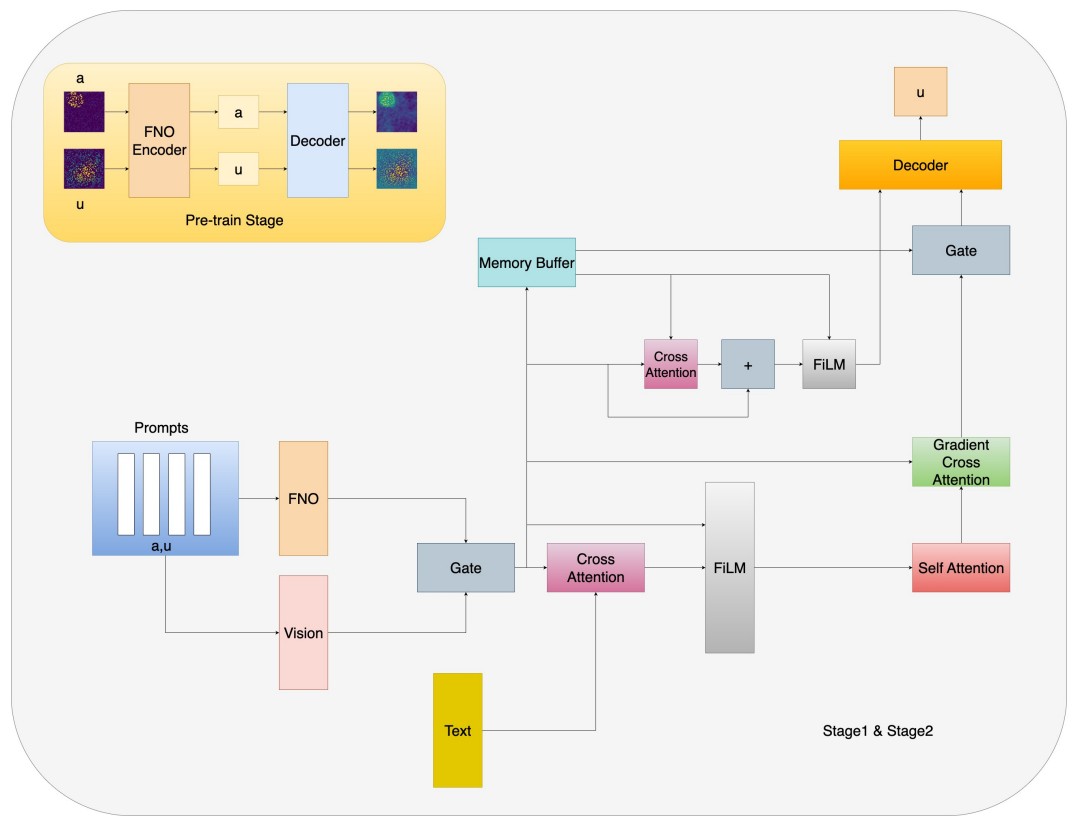

Figure 1: *Work Flow of MOFS*

| Dataset | Full | w/o pretrain | w/o text | w/o memory | w/o vision |
|---|---|---|---|---|---|
| DarcyFlow-100.0 | 0.0491 | 0.0589 | 0.0712 | 0.0530 | 0.0901 |
| DarcyFlow-0.1 | 0.1229 | 0.1818 | 0.1241 | 0.1250 | 0.1948 |
| IncomNS-100 | 0.0622 | 0.1122 | 0.0999 | 0.0952 | 0.0973 |
| IncomNS-101 | 0.0395 | 0.0650 | 0.0624 | 0.0635 | 0.0909 |
| IncomNS-0 | 0.0573 | 0.0733 | 0.1024 | 0.0937 | 0.0757 |

Figure 2: *Ablation study across different components of MOFS.*

Figure 3: *Visual comparison of fourier spectrum of latent representations for input $a$ and output $u$ in Pre-train Stage and Stage 2.*

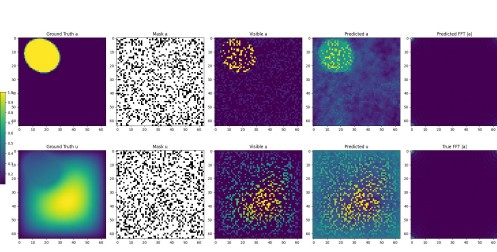

Figure 4: *Reconstruction of Masked Inputs in Pre-train Stage.*

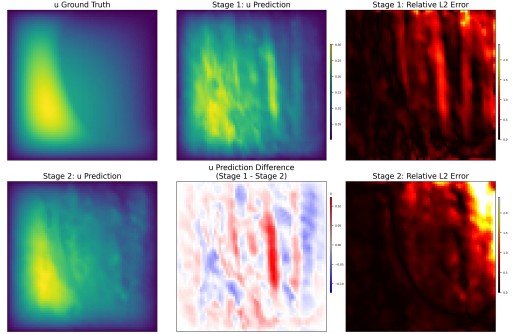

Figure 5: *Visual comparison of Stage 1 and Stage 2 predictions for $u$.*

| Datasets | DeepONet | FNO | UNet | Ours (MOFS) |
|---|---|---|---|---|
| DarcyFlow-100.0 | $0.1872 \pm 0.0127$ | $0.0854 \pm 0.0042$ | $0.2983 \pm 0.0028$ | $0.0491 \pm 0.0135$ |
| DarcyFlow-10.0 | $0.1795 \pm 0.0141$ | $0.0840 \pm 0.0043$ | $0.2974 \pm 0.0039$ | $0.0647 \pm 0.0108$ |
| DarcyFlow-1.0 | $0.2007 \pm 0.0126$ | $0.0820 \pm 0.0041$ | $0.3197 \pm 0.0032$ | $0.0325 \pm 0.0108$ |
| DarcyFlow-0.1 | $0.2530 \pm 0.0160$ | $0.1497 \pm 0.0042$ | $0.4290 \pm 0.0061$ | $0.1229 \pm 0.0190$ |
| DarcyFlow-0.01 | $0.4550 \pm 0.0210$ | $0.3148 \pm 0.0018$ | $0.7403 \pm 0.0122$ | $0.1331 \pm 0.0137$ |
| IncomNS-100 | $0.0750 \pm 0.0073$ | $0.0209 \pm 0.0008$ | $0.0654 \pm 0.0051$ | $0.0622 \pm 0.0023$ |
| IncomNS-101 | $0.1058 \pm 0.0018$ | $0.0189 \pm 0.0007$ | $0.0680 \pm 0.0020$ | $0.0395 \pm 0.0025$ |
| IncomNS-102 | $0.1134 \pm 0.0058$ | $0.0234 \pm 0.0008$ | $0.0976 \pm 0.0107$ | $0.0455 \pm 0.0006$ |
| IncomNS-10 | $0.2554 \pm 0.0132$ | $0.0272 \pm 0.0012$ | $0.1155 \pm 0.0090$ | $0.0671 \pm 0.0019$ |
| IncomNS-1 | $0.0846 \pm 0.0044$ | $0.0212 \pm 0.0013$ | $0.0850 \pm 0.0050$ | $0.0420 \pm 0.0024$ |
| IncomNS-0 | $0.1485 \pm 0.0078$ | $0.0207 \pm 0.0002$ | $0.0891 \pm 0.0043$ | $0.0573 \pm 0.0036$ |

Table 1: *Comparison of relative $\mathcal{L}_2$ error across datasets. The results are averaged over 3 runs in this paper.*

| Dataset | $J = 2$ | $J = 4$ |
|---|---|---|
| DarcyFlow-0.1 | $0.1288 \pm 0.0314$ | $0.1229 \pm 0.0190$ |
| IncomNS-101 | $0.0474 \pm 0.0028$ | $0.0437 \pm 0.0022$ |
| IncomNS-1 | $0.0659 \pm 0.0045$ | $0.0549 \pm 0.0050$ |
| IncomNS-0 | $0.0797 \pm 0.0059$ | $0.0597 \pm 0.0083$ |
| IncomNS-102 | $0.0623 \pm 0.0050$ | $0.0548 \pm 0.0020$ |

Table 2: *Ablation study for $J = 2$ and $J = 4$.*

| Dataset | Stage 1 | Stage 2 |
|---|---|---|
| DarcyFlow-0.1 | $0.1265 \pm 0.0092$ | $0.1229 \pm 0.0190$ |
| DarcyFlow-1.0 | $0.1026 \pm 0.1096$ | $0.0325 \pm 0.0108$ |
| DarcyFlow-0.01 | $0.2330 \pm 0.1106$ | $0.1332 \pm 0.0137$ |
| IncomNS-1 | $0.0595 \pm 0.0028$ | $0.0420 \pm 0.0024$ |
| IncomNS-0 | $0.0605 \pm 0.0036$ | $0.0573 \pm 0.0036$ |
| IncomNS-10 | $0.1035 \pm 0.0257$ | $0.0671 \pm 0.0019$ |

Table 3: *Comparison of relative $\mathcal{L}_2$ error between Stage 1 and Stage 2.*

**Results Analysis.** Table 1 reports the comparative performance of our proposed method, MOFS, against three strong baselines including DeepONet, FNO, and UNet on a set of PDE datasets. We report the relative $\mathcal{L}_2$ error averaged over 3 independent runs using different seeds. On DarcyFlow dataset, MOFS outperforms all baselines with notable margins, underscoring its robustness across different permeability scales. Table 3 presents the comparison of relative $\mathcal{L}_2$ error between Stage1 and Stage2. Figure 3 compares the fourier spectrum of latent representations for input $a$ and output $u$ in Pre-train Stage and Stage 2. Figure 4 presents pre-train stage results, where the model is trained to reconstruct masked inputs for both the coefficient field $a$ and solution field $u$. To assess the efficacy of our staged training strategy, we visualize and compare the predicted solution fields $u$ from Stage 1 and Stage 2 in Figure 5. Figure 2 presents an ablation study assessing the contribution of different components in the MOFS framework. The full model consistently achieves the lowest relative error. Table 2 reports the ablation results for different $J$. The detailed analysis of Figure 3, 4, 5 are shown in Appendix.

## 6 CONCLUSION

We propose MOFS: a unified multimodal framework for **m**ulti-**o**perator **f**ew-**s**hot learning that enables generalization across PDE families using limited demonstrations. Our approach leverages a shared FNO encoder, pretrained via spatial-frequency self-supervision, alongside semantically grounded text embeddings and memory-augmented prompting. Through gated and attention-based fusion of visual, spectral, and textual modalities, our model constructs operator-aware representations that support prompt-based inference. To facilitate robust generalization, we introduce a two-stage training strategy: supervised prompt-conditioned learning followed by end-to-end contrastive fine-tuning. Experimental results across PDE benchmarks, including Darcy Flow and Incompressible Navier Stokes variants, demonstrate that our method outperforms traditional neural operator baselines in few-shot scenarios. Extensive ablation studies confirm the individual and joint contributions of each modality and training component. Our work lays the foundation for future research on universal operator learning by bridging techniques from physics-informed modeling, contrastive representation learning, and multimodal prompting.

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
