# OpenReview forum: "Multi-Operator Few-Shot Learning for Generalization Across PDE Families"
_ICLR.cc/2026/Conference — Submitted to ICLR 2026_

### Official Review · Reviewer_5i8H · 2025-10-19

**Soundness:** 1
**Presentation:** 2
**Contribution:** 1
**Rating:** 0
**Confidence:** 5

**Summary:**

This paper proposes MOFS, a multimodal framework for few-shot operator learning across PDE families. The method combines (i) self-supervised pretraining of a shared FNO encoder with spatial and frequency reconstruction, (ii) text-conditioned operator embeddings derived from statistical summaries of PDE fields, and (iii) a memory-augmented prompting mechanism that retrieves prior PDE examples to enhance inference. The training pipeline involves two stages: prompt-conditioned supervised learning and contrastive fine-tuning. Empirical evaluations on Darcy Flow and Incompressible Navier–Stokes benchmarks are presented, with claims of superior few-shot generalization compared to standard neural operator baselines such as FNO, DeepONet, and UNet.

**Strengths:**

- Clarity of the high-level idea: The paper is decently organized and describes the main components of MOFS: spatial-frequency pretraining, textual embeddings, and memory-augmented multimodal fusion. The architecture and training pipeline are broken down step by step.

- Ambitious goal: Tackling cross-operator few-shot generalization is a challenging and relevant direction in operator learning, where standard neural operator methods typically overfit to a single PDE family.

- Some empirical performance gains: Reported results on Darcy Flow and Incompressible Navier–Stokes show lower relative L2 error compared to baseline methods in a few-shot setting.

- Ablation studies: The authors attempt to evaluate contributions of different components (e.g., without text, vision, memory, pretraining).

**Weaknesses:**

The technical novelty and empirical evidence fall short in several critical areas:

- Lack of genuine novelty:
Most components of the proposed framework are borrowed or adapted from existing multimodal and meta-learning architectures (e.g., cross-modal attention, CLIP-style text embeddings, memory retrieval). There is little actual innovation at the algorithmic or theoretical level. The “integration” of standard components does not by itself constitute a significant research contribution.

- Weak baselines and missing comparisons:
The experiments only compare against fairly basic baselines (FNO, DeepONet, UNet). These are insufficient to support the paper’s strong claims of “new foundation” for operator generalization. Crucially, the method should be compared against stronger and more recent approaches such as: (1) Koopman neural operator as a mesh-free solver of non-linear partial differential equations, (2) Solving High-Dimensional PDEs with Latent Spectral Models. Without these baselines, it’s unclear whether MOFS provides any real improvement over the state of the art.

- Overly complex architecture with limited justification:
The method introduces a very large number of components (pretraining objectives, text embeddings, cross-attention, memory retrieval, soft prompts, multiple loss functions). However, the ablation study is superficial — it shows that “removing components hurts performance,” but does not explain why or which components are essential versus redundant. This creates the impression of “architecture inflation” rather than targeted innovation.

- Very limited evaluation scope:
The evaluation is confined to Darcy Flow and Navier–Stokes variants with fairly simple variations. This is insufficient to substantiate the strong generalization claims. No high-dimensional PDEs, irregular domains, or more complex physics are tested.

- No robustness or efficiency analysis:
Given the model’s complexity, it likely has high computational overhead. The paper does not provide inference latency, memory footprint, or training cost. This is critical for a method claiming to improve “few-shot efficiency.”

- Lack of theoretical grounding:
Unlike some prior works on operator generalization and meta-PDE learning, the paper does not provide any theoretical insight into why multimodal prompting should yield improved generalization. The method is essentially heuristic.

- Weak justification for text embeddings:
The paper uses simple field statistics to generate textual descriptions. It’s not clear how this weak textual signal actually helps operator learning. There is no ablation isolating textual conditioning’s effect in a controlled way beyond a single table entry.

- Writing tone overstates contributions:
The abstract and introduction repeatedly claim “first framework,” “universal foundation,” etc., without sufficient empirical or theoretical support.

**Questions:**

- Missing baselines:
Please compare against the following state-of-the-art PDE operator learning methods: (1) Koopman neural operator as a mesh-free solver of non-linear partial differential equations, (2) Solving High-Dimensional PDEs with Latent Spectral Models. These are essential baselines to evaluate whether MOFS offers any meaningful improvement.

- Component justification:
Why is textual conditioning necessary? What specific structure is learned from the textual descriptions that is not already captured in the numerical fields?

- Complexity analysis:
Please provide a breakdown of computational cost — both during training and inference — compared to a standard FNO. How does the multimodal fusion and memory retrieval affect scalability?

- Generalization beyond toy PDEs:
How does the method perform on more challenging PDEs (e.g., advection–diffusion, wave equations, or higher-dimensional problems)? Are there failure cases?

- Ablation clarity:
The current ablations are too coarse. Please conduct more fine-grained analyses (e.g., only pretraining vs. only memory vs. only text, different prompt lengths, memory size sensitivity, etc.).

- Unclear effect of memory buffer:
How large is the memory, how is it updated, and how sensitive is the model to its size and quality? What happens if the retrieved prompts are noisy or irrelevant?

---

> ### Author Response · Authors · 2025-11-27
>
> > Missing baselines: Please compare against the following state-of-the-art PDE operator learning methods: (1) Koopman neural operator as a mesh-free solver of non-linear partial differential equations, (2) Solving High-Dimensional PDEs with Latent Spectral Models. These are essential baselines to evaluate whether MOFS offers any meaningful improvement.
>
> R1: Our dataset is on standard grid, so we don't consider mesh-free solver of non-linear partial differential equations here.
>
> And our model does not focus on high-dimensional PDEs,and our problem formulation is defined in Line 111-127.
>
>
>
> > Component justification: Why is textual conditioning necessary? What specific structure is learned from the textual descriptions that is not already captured in the numerical fields?
>
> R2: Textual conditioning can improve the performance. And the embedding from textual descriptions using language model is different from other kinds of modals. And this is also consistent with multi-modal model proposed in the paper.
>
>
> > Complexity analysis: Please provide a breakdown of computational cost — both during training and inference — compared to a standard FNO.
>
> R3: This inofrmation can be found in Appendix in Line 662.
>
> The parameter count of MOFS is 32,266,030. GPU is NVIDIA GeForce RTX 3090. Average time cost for MOFS is about 3,035 seconds.
>
>
> > Generalization beyond toy PDEs: How does the method perform on more challenging PDEs (e.g., advection–diffusion, wave equations, or higher-dimensional problems)?
>
> R4: We conduct experiments on the datasets listed in the paper. If we generalize it to a real-world dataset or other challenging PDEs, the results need to be confirmed.
>
>
> > Ablation clarity: The current ablations are too coarse. Please conduct more fine-grained analyses (e.g., only pretraining vs. only memory vs. only text, different prompt lengths, memory size sensitivity, etc.).
>
> R5: Our abaltion study results are shown on Table 2, Table 3, Figure 2. We choose different and various directions to test each component. And the results have proven the effectiveness of each part.
>
> > Unclear effect of memory buffer: How large is the memory, how is it updated, and how sensitive is the model to its size and quality? What happens if the retrieved prompts are noisy or irrelevant?
>
> R6: The size of memory buffer is 1024. In Stage 1 and Stage 2, we start to choose qualified FNO embedding for a and u to memory buffer and update memory buffer continuously.
>
> Although memory buffer is large, when we train the model, we only choose top-k pairs to our model, so top-k is the key for performance here instaed of size of memory buffer.
>
> Actually, we design many mechanisms to choose better FNO embedding for a and u to improve results.
>
> So there is no worry for noisy or irrelevant retrieved prompts.

---

### Official Review · Reviewer_4cLZ · 2025-10-25

**Soundness:** 2
**Presentation:** 2
**Contribution:** 2
**Rating:** 2
**Confidence:** 3

**Summary:**

The paper introduces MOFS (Multi-Operator Few-Shot Learning), a multimodal neural framework for learning across families of PDE operators. The key idea is to generalize to unseen PDEs using only a few demonstration pairs by combining (i) frequency-aware FNO pretraining, (ii) text-conditioned embeddings derived from field statistics via BERT, and (iii) a memory-augmented prompting mechanism for cross-modal attention. The authors adopt a two-stage training scheme: supervised few-shot learning followed by contrastive fine-tuning to align latent representations across modalities. Experiments on variants of Darcy Flow and Incompressible Navier-Stokes equations show that MOFS achieves lower relative $L^2$ errors than FNO, DeepONet, and UNet baselines.

**Strengths:**

1. The paper is easy to follow.

2. The attempt to use textual statistics as priors and align them with spectral/visual features is interesting and novel within the operator-learning literature.

**Weaknesses:**

1. Lack of conceptual novelty.
    -    Most components—FNO encoder, contrastive learning, text conditioning, memory-based prompting—are directly adapted from existing architectures (e.g., FNO, CLIP, Flamingo) with minimal innovation specific to operator learning. The paper does not articulate why multimodal fusion or text embeddings are theoretically beneficial for PDEs beyond empirical combination.

2. Weak empirical reuslts.
    - Although large and diverse public benchmarks such as PDEBench, PDEArena, and Well datasets are available, the experiments only use small subsets (Darcy Flow and simple Navier–Stokes variants) with low resolution and few configurations.
    - The baselines are outdated. More relevant and competitive comparisons such as UFNO, AFNO, CViT, and recent PDE foundation models including MPP, Poseidon, DPOT, PDEformer are missing. Given the rapid progress in neural operator research, the presented experiments do not provide convincing evidence of state-of-the-art performance

3. Incomplete experimental details
    - The paper lacks essential information such as the explicit PDE formulations, discretization resolutions, hyperparameter settings (e.g., learning rate schedules, training epochs, optimizers), model sizes, and computational costs. Without these critical details, it is impossible to evaluate the fairness of comparisons or ensure reproducibility of the reported results.

**Questions:**

1. Dataset and setup:
   - What specific PDE equations and boundary conditions are used for each dataset? What are the spatial resolutions and data size for the pretraining stage?
   - What are the exact hyperparameter settings (learning rate schedules, optimizers, batch size, number of epochs, etc.) and model sizes?

2. How are the text descriptions generated for unseen PDEs in few-shot evaluation? Given the small number of samples, how stable or reliable are the computed statistics?

3. How large is the memory buffer in practice, and how is it maintained during training and inference? Does memory retrieval improve results beyond simply enlarging model capacity?

---

> ### Author Response · Authors · 2025-11-27
>
> > What specific PDE equations and boundary conditions are used for each dataset? What are the spatial resolutions and data size for the pretraining stage?
>
>
> R1: The dataset is from PDEBench and we explain this detail in Line 365.
>
>
>
>
> > What are the exact hyperparameter settings (learning rate schedules, optimizers, batch size, number of epochs, etc.) and model sizes?
>
>
> R2:
>
> | Parameter                     | Value              |
> |------------------------------|--------------------|
> | mask ratio                   | 0.25               |
> | target size                  | 64                 |
> | width                        | 128                |
> | number of modes              | 12                 |
> | schedule for pre-train stage | LambdaLR           |
> | optimizer for pre-train stage| AdamW              |
> | schedule for stage 1         | CosineAnnealingLR  |
> | optimizer for stage 1        | AdamW              |
> | schedule for stage 2         | CosineAnnealingLR  |
> | optimizer for stage 2        | AdamW              |
>
> The number of epochs is dynamic in our model. We set standards to contiune in one stage. If there is no improvement for performance after 80 epochs, the process will continue to the next stage.
>
> > How are the text descriptions generated for unseen PDEs in few-shot evaluation? Given the small number of samples, how stable or reliable are the computed statistics?
>
> R3: In our model, we only condition on text descriptions generated by seen PDEs. So there is no worry for unseen PDEs in few-shot evaluation.
>
>
>
>
>
> > How large is the memory buffer in practice, and how is it maintained during training and inference? Does memory retrieval improve results beyond simply enlarging model capacity?
>
> R4: The size of memory buffer is 1024. In Stage 1 and Stage 2, we start to choose qualified FNO embedding for a and u to memory buffer and update memory buffer continuously.
>
> Although memory buffer is large, when we train the model, we only choose top-k pairs to our model, so top-k is the key for performance here instaed of size of memory buffer.
>
> Actually, we design many mechanisms to choose better FNO embedding for a and u to improve results.

---

### Official Review · Reviewer_XqoS · 2025-10-31

**Soundness:** 1
**Presentation:** 1
**Contribution:** 2
**Rating:** 2
**Confidence:** 4

**Summary:**

The paper gives a new method for few-shot learning of PDE surrogates via operator networks. It first consists of a pretraining reconstruction stage (spatial and spectral), followed by two stages of supervised operator learning. The first stage does few shot learning by using the inputs, embeddings, similarly retrieved "prompts", and a text embedding of a language description of the statistics of the PDE data.  The second stage minimizes the spatial prediction loss again, but now also with contrastive terms encouraging similar embeddings within PDE datasets and a term encouraging sufficient spread of the embeddings in a memory buffer.  Experiments are run on a few selections from PDEBench.

**Strengths:**

Exploring operator learning techniques which can generalize via few-shot learning across different PDEs is an important area of research.  Using pretraining approaches for such methods is a promising way forward to more efficient and broadly applicable models for the fast numerical surrogate solutions for PDEs.

**Weaknesses:**

The paper is unclear overall in its presentation of the method and gives little motivation for each of the many components.

The use of a language model to encode the statistics of the samples for each PDE dataset feels particularly unmotivated.  What is the additional benefit of a natural language representation of these scalar statistics compared to dealing with them directly?  The paper writes that this captures both "physical statistics and linguistic priors."  It is unclear what a linguistic prior means here when the descriptive sentence is entirely composed of a label of the PDE dataset and its numerical statistics.

There are almost no details given around the experiment implementations, making any attempted reproduction of these results nearly impossible.

**Questions:**

1. In the pretraining phase, why is only the spatial component masked?

2. Why does the frequency loss for pretraining only look to reconstruct the input frequencies and not the output function's frequencies as well?

3. What are the settings of all method parameters used for the experiments?  E.g. masking ratio, architecture sizes (lengths, widths, number of modes, etc).  Which layers are frozen during pretraining?  What were the corresponding hyperparameters/architecture choices for the models compared against?

4. The paper claims "physics supervision" is used in the second few-shot stage, is this referring to the L2 supervised loss of the output of the model with the ground truth field?  Typically this would instead refer to the satisfaction of certain physical constraints, such as the PDE operator itself or some other conservation law.

---

> ### Author Response · Authors · 2025-11-27
>
> > In the pretraining phase, why is only the spatial component masked?
>
> R1: We randomly mask some portion of a and u on grid. And then we use our method proposed in 4.1 Multi-Task Pretraining with Spatial and Frequency Reconstruction to pre-train. Specifically, when we decode spatial component and frequency component, the input of them is both the masked FNO embedding of a and u. So we mask them on spatial direction, and reconstruct spatial part and frequency part using masked FNO embedding of a and u. The details for this part are explained in the paper in Line 137-160.
>
>
> > Why does the frequency loss for pretraining only look to reconstruct the input frequencies and not the output function's frequencies as well?
>
> R2: The motivation here is as follows. We are doing operator learning, a mapping from a to u. So we aim to predict u given a. We use masked FNO embedding of a and u as frequency decoder's input to reconstruct a's frequency. The use of u here is to strengthen decoder's information. Our final goal is to use the information from a to reconstruct a, and catch the inner laws of a to predict u better.
>
>
> > What are the settings of all method parameters used for the experiments? E.g. masking ratio, architecture sizes (lengths, widths, number of modes, etc).
>
> R3:
>
> | parameter        | value |
> |-----------------|-------|
> | mask ratio      | 0.25  |
> | target size     | 64    |
> | width           | 128   |
> | number of modes | 12    |
>
> > Which layers are frozen during pretraining?
>
> R4: Please see Algorithm 1 in the paper. In pretraining stage, we aim to train FNO encoder which is prepared for Stage 1 and Stage 2. No layers are frozen during pretraining.
>
>
>
>
> > The paper claims "physics supervision" is used in the second few-shot stage, is this referring to the L2 supervised loss of the output of the model with the ground truth field? Typically this would instead refer to the satisfaction of certain physical constraints, such as the PDE operator itself or some other conservation law.
>
> R4: Yes, I agree with you.

---

### Official Review · Reviewer_7Nu4 · 2025-11-01

**Soundness:** 3
**Presentation:** 3
**Contribution:** 3
**Rating:** 6
**Confidence:** 2

**Summary:**

In this paper, the authors introduce MOFS, a multi operator for few-shot learning. Aiming for generalizations to unseen PDE operators. Leveraging pre-training using a combination of spatial and frequency losses, a few-shot fine-tuning and an end-to-end contrastive learning phase based on a multimodal approach involving text-conditioned embedding and memory embedded prompting, this model enjoys high performance when compared to other classic neural operator architectures such as FNO or DeepOnet.

**Strengths:**

-	Very innovative multi-modal approach
-	Strong performance is achieved on the benchmark PDEs considered.

**Weaknesses:**

-	The comparisons with other models, such as DeepOnet or FNO, are a bit unclear; do we consider a similar number of  parameters for these models ?
-	Not much infration is given regarding how expensive the various phases of training are, in particular the contrastive learning.
-	The text conditioning is hard coded, and could possibly hinder the generalization of the method to more complex PDEs.

**Questions:**

-	Can the format of the text used generalize to more complex PDEs ? would we need to consider more complex statistics ?
-	What if the PDE name and the various statistical data where embedded in a different way (say, some labels); and the model then conditioned on them.would the performance be diminished ?
-        How expensive is the whole training procedure in comparison to that of a classic neural operator model ?

---

> ### Author Response · Authors · 2025-11-27
>
> We thank the reviewer for the constructive comments.
>
> > Can the format of the text used generalize to more complex PDEs ? would we need to consider more complex statistics ?
>
> R1: For different and more complex PDEs, we use the same framework and method proposed in the paper to generate its corresponding text embedding. In our current method, we consdier some general statistics listed in the paper to describe the PDEs and it has achieved the goal.
>
>
>
> > What if the PDE name and the various statistical data where embedded in a different way (say, some labels); and the model then conditioned on them.would the performance be diminished ?
>
> R2: In our current method, we use a direct way to embed PDE name and the various statistical data. We don't transfer these information to labels and then use labels to embed. It's a curious direction to think about embedding. Actually, I want to employ language model here, so I generate its descriptive sentence first and then use encoder to get its corresponding embedding. This is the motivation for this part.
>
> > How expensive is the whole training procedure in comparison to that of a classic neural operator model ?
>
>
> R3: Our method's complexity is listed in Appendix. The parameter count of MOFS is 32,266,030. GPU is NVIDIA GeForce RTX 3090. Average time cost for MOFS is about 3,035 seconds.

---

### Meta-Review · Area_Chair_HcXV · 2026-01-03

**Summary:**

This paper proposed a unified multimodal framework MOFS for multi-operator few-shot learning. The method integrates (1) multi-task self-supervised pretraining of a shared FNO encoder; (2) text-conditioned operator embeddings derived from summaries of input-output fields; and (3) memory-augmented multimodal prompting with gated fusion and cross-modal gradient-based attention. Expreiments on PDE benchmarks show the good performance of the method.

The reviewers have raised several major concerns of this paper.
- Limited novelty: Reviewer 4cLZ and 5i8H pointed out the novelty issue. Most components of the proposed framework are from existing architectures.
- Weak experimental results: Reviewer 4cLZ and 5i8H pointed out the weak experimental results and lack of baselines in experiments. Therefore the effectiveness of the method is not convincing.
- Unclear experimental details: Reviewer XqoS and Reviewer 4cLZ pointed out that the experimental details are not clear in the context.
- Text conditioning is not reasonable: Reviewer 7Nu4, XqoS, and 5i8H pointed out that using text condition is not well-motivated.

Besides, there are many individual concerns raised by the reviewers.

**Reviewer Concerns:**

The authors provided a short rebuttal to address the reviewer concerns. The reviewers did not engage in the discussion. After reading the rebuttal, I think most of the concerns were not addressed, especially the important ones like limited novelty and weak results.

**Reviewer Scores:**

I think the reviewers would not change their scores after rebuttal.

---

### Decision · Program_Chairs · 2026-01-26

Reject